# ACF: An Armed CCTV Footage Dataset for Enhancing Weapon Detection

**DOI:** 10.3390/s22197158

**Published:** 2022-09-21

**Authors:** Narit Hnoohom, Pitchaya Chotivatunyu, Anuchit Jitpattanakul

**Affiliations:** 1Image Information and Intelligence Laboratory, Department of Computer Engineering, Faculty of Engineering, Mahidol University, Nakhon Pathom 73170, Thailand; 2Department of Mathematics, Faculty of Applied Science, King Mongkut’s University of Technology North Bangkok, Bangkok 10800, Thailand; 3Intelligent and Nonlinear Dynamic Innovations Research Center, Science and Technology Research Institute, King Mongkut’s University of Technology North Bangkok, Bangkok 10800, Thailand

**Keywords:** weapon detection, image tiling, deep learning, armed CCTV footage dataset, surveillance system

## Abstract

Thailand, like other countries worldwide, has experienced instability in recent years. If current trends continue, the number of crimes endangering people or property will expand. Closed-circuit television (CCTV) technology is now commonly utilized for surveillance and monitoring to ensure people’s safety. A weapon detection system can help police officers with limited staff minimize their workload through on-screen surveillance. Since CCTV footage captures the entire incident scenario, weapon detection becomes challenging due to the small weapon objects in the footage. Due to public datasets providing inadequate information on our interested scope of CCTV image’s weapon detection, an Armed CCTV Footage (ACF) dataset, the self-collected mockup CCTV footage of pedestrians armed with pistols and knives, was collected for different scenarios. This study aimed to present an image tilling-based deep learning for small weapon object detection. The experiments were conducted on a public benchmark dataset (Mock Attack) to evaluate the detection performance. The proposed tilling approach achieved a significantly better mAP of 10.22 times. The image tiling approach was used to train different object detection models to analyze the improvement. On SSD MobileNet V2, the tiling ACF Dataset achieved an mAP of 0.758 on the pistol and knife evaluation. The proposed method for enhancing small weapon detection by using the tiling approach with our ACF Dataset can significantly enhance the performance of weapon detection.

## 1. Introduction

The number of crimes perpetrated as a result of fraud or robbery has risen steadily year after year. The availability of evidence that might be utilized to arrest and punish offenders would strengthen the judicial system. Such evidence may be needed to accurately identify the offenders and specific aspects of the crime scene. Closed-circuit television (CCTV) systems are a type of recording equipment that can be utilized for security purposes. This is made feasible by surveillance with cameras that capture crime events. The reliability of CCTV imaging is a significant advantage. However, helpful evidence typically depends on video quality, which must capture high-resolution images to unmistakably identify criminals or information such as human faces, license plates of offender vehicles, or other evidence that appears on the footage. CCTV cameras have become essential in addressing security issues and are regarded as one of the most critical security needs [1]. Initially, CCTV cameras were developed to control undesirable events or behavior in cities through public administration-related activities for control and prevention. Almost two million CCTV cameras have been installed in the United Kingdom, with 400,000 CCTVs primarily deployed as surveillance systems.

Thailand is also faced with a wide range of illegalities, such as offenses against property and life and unpleasant conduct that causes others to feel frightened. According to the Royal Thai Police statistical data of crimes recorded in 2020 [2], there were 14,459 criminal offenses reported in Thailand. The officers apprehended and sentenced 13,645 perpetrators, or 94.37 percent, while 5.63 percent of the offenders reported had not been apprehended. In Bangkok, there were 2119 criminal actions committed. The police department apprehended 1897 perpetrators, 89.52 percent of all offenders, while nearly 10 percent have not been apprehended. Consequently, 52.20 percent of Bangkok residents felt insecure, according to a poll of their views about crime [2]. Furthermore, there is the instance of premeditated murder in Korat or Nakhon Ratchasima province [3]. The culprit drove a vehicle outfitted with military weaponry and shot at people without regard for their safety. The criminal siege of Korat’s Terminal 21 resulted in 29 people killed and more than 58 people wounded.

To maintain public safety, it is important to leverage technology to aid surveillance systems. Several provinces, including Bangkok and Phuket, have installed CCTV systems. These systems still have limitations, such as requiring personnel to continually monitor the displays. This could lead to shortcomings in particular events or circumstances due to human limitations, such as inexplicable monitoring. Due to large crowds and spectacles, an officer may be unable to collect valid evidence to implicate the culprit, or the officer may be unable to act when an occurrence happens. Artificial intelligence technology may detect suspicious activity in a CCTV system, such as screening citizens for carrying weapons, monitoring a person’s movements, and screening for escape routes to track offenders.

The problem can be solved by combining weapon detection and CCTV cameras. The weapon detection system can reveal the behavior of a suspect, reducing the risk of harm to those who are not afraid of the law and justice processes, as well as protecting the population against harm. Surveillance cameras installed at strategic points throughout the city can be used to identify a suspect, and the system analyzes real-time video streams in the IoT cloud. It analyzes anomalies of the video streaming data with the results of abnormal detection through the IoT monitor system to alert the nearest local police station with a weapon detection IoT system installation. This can be achieved by employing CCTVs to detect video footage in the installation areas to determine the location of an incident.

The objectives of this study were to develop a weapon detection model for CCTV camera footage and to enhance the efficiency of object detection models for detecting weapons. The data used in this research were self-collected on real CCTV cameras with setup scenarios. The concerned weapons for detection are pistols and knives, as these types of weapons are primarily used in crimes. The transfer learning approach of ImageNet [4] and Common Objects in Context (COCO) [5] pre-trained object detection models were used to train object detection in this study.

Detecting weapons on CCTV footage is challenging because the images’ weapons are far away and are not visible. The characteristics of small weapon images are limited and challenging to identify. There were several problems in detecting training objects, which are listed below:The first and most critical issue was that the data are fed to a Convolution Neural Network (CNN) for learning features to achieve classification and detection tasks. For the detection of weapons on CCTV footage, there was no standard dataset available.The public datasets primarily available did not cover our interesting tasks, such as a pistol images dataset containing a full-size pistol in images for the classification task. In most of the datasets, the image quality was poor and unsymmetrical.Manually collecting data was a time-consuming task. Furthermore, labeling the collected dataset was complex because all data must be categorized manually.As mentioned, the objects in the images extracted from CCTV footage were small. Therefore, a preprocessing method is needed to enhance the efficiency of the object detection models.

The key contributions of this study were a thorough analysis of weapon detection in CCTV footage and the identification of the most suitable and appropriate deep learning object detection for weapon detection in CCTV footage.

The collection of new datasets for appropriate weapon detection training data, in which we collected an Armed CCTV Footage (ACF) dataset of 8319 images. The dataset contained CCTV footage of a pedestrian carrying a weapon in different scenes and armed postures. The ACF Dataset was collected in indoor and outdoor scenarios to leverage the efficiency of weapon detection on actual CCTV footage.The research implemented the image tiling method to enhance the object detector efficiency for small objects. The image tiling method will help enhance weapon detection on the small image object region.

The entirety of the paper is arranged as follows: Section 2 discusses a literature review and public dataset availability; Section 3, the processes of the self-collected dataset, the ACF Dataset. Section 4 discusses the experiments conducted to improve the efficiency of the image tiling method in detecting weapons, while Section 5 evaluates the experimental results using public and ACF Datasets on weapon detection. In Section 6, the methods and results of the experiments are deliberated. Finally, Section 7 presents the conclusions and future work.

## 2. Literature Review

A weapon detector is a neural network architecture that detects armed individuals attempting to commit a crime. The weapon detection was designed to work with pre-existing CCTV systems, whether public or private, to help notify local security and identify possible crimes. The authors believe that implementing a weapon detector in existing CCTV systems will help reduce the number of officers required for CCTV coverage while enhancing the advantages of CCTVs.

### 2.1. Object Detection Approach Overview

In the early revolution of machine learning, image processing methods utilized the field of vision to overcome the on-domain problems. The development of image processing with machine learning techniques has been utilized to detect weapons based on object characteristics and patterns. A color-based segmentation and interest point detector hybrid approach has been used for the automated identification of weapons [6,7] to detect objects and to analyze their characteristics for matching with pistol descriptions. In addition, the Speeded Up Robust Features (SURF) interest point detector is widely used to identify items inside an image; it employs a blob or edges to locate the weapons in the image. Tiwari et al. [6] proposed and evaluated a model that uses color-based segmentation; the system performs substantially better when photos include diverse attributes. Matching determines the similarity score between the gun and blob stored descriptors.

While deep learning is gaining popularity as a solution to artificial intelligence challenges. The algorithm is capable of interpreting data across the network. Deep learning for computer vision has led to several interesting advances, including self-driving cars [8] and Alpha Go [9], which can compete with World Go Champions. In the field of computer vision, deep learning frameworks, such as TensorFlow [10], Caffe [11], Pytorch [12], and Onnx [13], have provided friendly access for developers as well as flexibility and interoperability. These frameworks support scientific computing that offers broad support for machine learning algorithms such as image classification, text recognition, or object detection, as the object detection method is developed by enhancing the image classification to localize objects along with identifying the type of object [14,15,16]. The CNN-based object detector consists of two types: single-shot (one-stage) detector and region-based (two-stage) detectors. The single-shot detectors prioritize speed over precision and attempt to simultaneously predict both the bounding box and the class, as shown in Figure 1. The single-shot multi-Box detector (SSD) [17] and all variants of You Only Look Once (YOLO) [18,19,20] are examples of single-shot detectors. Some widely used studies [16,21,22,23,24,25,26] have focused on attention mechanisms to collect more helpful characteristics for locating objects.

The region-based detectors initiate by assembling a sparse collection of regions in the image where objects are identified. The second step categorizes and refines the projected position of each object proposal as shown in Figure 2. The region-based convolutional neural network (R-CNN) [27] was an early version of CNNs that achieved significant accuracy gains. Later studies, such as Fast R-CNN [28], and Faster R-CNN [29] propose improvements to R-CNN to enhance its speed.

The Feature Pyramid Network (FPN) is commonly used in R-CNN, which is a feature extractor that was created with accuracy, and Lin et al. [30] attempted to retrieve better features by splitting these into multiple levels in order to recognize objects of various sizes and fully utilize the output feature maps of succeeding backbone layers. A few works [31,32,33,34] leverage FPN as the backbone of their multi-level feature pyramid.

### 2.2. Weapon Detection Public Datasets

The CCTV footage for surveillance can be included in various systems such as traffic systems, pedestrian analysis, or transportation services maintenance. CCTV cameras are mostly installed by facing the cameras out from a building; the footage often includes roads and pedestrians. Traffic volume detection [35,36] is implemented using the deep learning object detection algorithm, predicting the future volume of traffic. For the safety of the population, the CCTV system can be used for pedestrian detection [37], pedestrian’s behavior, and human activity [38,39,40] to enhance the surveillance system for accidents and risks threatening the population. For protecting innocent pedestrians from crimes that may happen, weapon detection needs to be able to report possible threats.

Carrobles et al. [41] and Bhatti et al. [42] introduced a Faster Region-based Convolutional Neural Network (Faster R-CNN) for surveillance to increase the efficiency and utility of CCTV cameras in the weapon detection task. The study employed Faster R-CNN technology to create a gun and knife detection system, where the authors used a public dataset that varied the size of the images and weapon objects. The data for crime intention were collected on images of crime with the intention of using guns. The following public image datasets are from the Weapon detection dataset [43] and Weapon detection system [44].

As an open topic for enhancing small item recognition and data collecting, González et al. [45] presented real-time gun detection in surveillance cameras. The authors created synthetic datasets using the Unity gaming engine and included CCTV footage gained from a simulated attack on the University of Seville. For its use in real-time CCTV, the study utilized a Faster R-CNN architecture based on the FPN. A combination of initially training synthetic pictures and then actual photos was the best strategy for improving the precision of detecting tiny things from tests, due to the CCTV footage of criminals carrying firearms in a distant view in such a small pixel region. The image tiling approach is commonly utilized in the identification of tiny objects. To ease tiny object problems, it is necessary to reduce the influence of picture down sampling, which is a frequent strategy employed during training. The image tiling method can be used with a variety of tile sizes, overlap amounts, and aggregation processes. The image is split into smaller images using overlapping tiles [46,47,48]. Overlapping tiles crop the lower quality pictures from the source image. Regardless of object size, each tile represents a fresh picture in which the ground truth object positions are set.

To keep computational needs low [49,50,51], neural networks frequently operate on small picture sizes, making it difficult to recognize microscopic objects. The use of tiling to process can greatly enhance detection performance while still preserving performance comparable to CNNs that scale and analyze a single image. Huang et al. [52] discovered that the use of zero-padding and stride convolutions resulted in prediction heterogeneity around the tile border, along with interpretation variability in the results. Unel et al. [53] used an overlapping tiles picture approach that greatly enhances performance, increasing the MAP (IoU: 0.5) from 11 percent to 36 percent. The precision of tiling objects was raised by 2.5× and 5×, respectively. The tiling approach is more successful on small items, as it also improves detection on medium-sized items. Daye [54] mentioned that instead of resizing the image to network size, image tilling decomposes an image into overlap or non-overlap patches, and each patch is fed to the network. The image can retain the original quality and details of features. As a result, high ratio scaling has no effect on the image, and small objects retain their sizes. Image tilling, however, can lengthen inference time and add extra post-processing overhead for merging and refining predictions.

For the research of weapon detection for a CCTV surveillance system, the weapon images datasets available to the public are limited. However, some datasets provide an annotation along with the dataset, but the weapon images contained in the dataset often are images from internet sources such as commercial-use images, weapon video content, movies or even cartoons. Only few of them contain CCTV footage with armed pedestrians as we required. We found three publicly available datasets that had some similarities in our interested domain, which are summarized in Table 1.

The Weapon detection dataset [43] comprises a total of 3000 pistol images. The images contained in this dataset contain pistol and rifle objects that are clearly visible. Some images contain a person holding a weapon straight to the camera. The Weapon detection system dataset [44] is composed of pistol images and rifle images from the Internet Movie Firearms Database (IMFDB) dataset and some CCTV footage comprising 4940 images. The images have various sizes from 99 × 93 pixels to 6016 × 4016 pixels with different ranges of vision and composition. The images from the IMFDB dataset are mostly from wars or crime-related movies, which mostly used a rifle-type of weapon, followed by pistols and shotguns. In these types of movies, the color tone is often dark and contains a lot of smoke, which can reduce the quality of weapon characteristics in an image.

As different environments might bring effects to model performance, Feris et al. [56] obtained datasets from the Maryland Department of Transportation website, by gathering a collection of roughly 40 short video clips, according to the following weather conditions: cloudy, sunny, snow, late afternoon, evening, and shadows. They chose roughly 50 images at random for each environmental condition and labeled them by drawing bounding boxes along the vehicles. This group of annotated images serves as the assessment standard. The detector is biased toward “Cloudy” conditions. Due to variables such as poor contrast, headlights, and specular reflections, performance lowers significantly for the “shadow” and “evening” situations. Due to appearance changes, performance is also impaired under the “snow”, “sunny”, and “late afternoon” circumstances.

The mock attack dataset collection from González et al. [45] uses CCTV footage that we are interested in developing for the system. The dataset is available on GitHub “Deepknowledge-US/US-Real-time-gun-detection-in-CCTV-An-open-problem-dataset” [55]. The mock attack dataset comprises full HD images with dimensions of 1920 × 1080 pixels and consists of 5149 images of mock attack scenarios, but only the weapon labels consist of 2722 labels of three weapon types: handguns, short rifles, and knives. The dataset was collected from three CCTV cameras located in different corridors and an entrance. The weapon images available had varied sizes and characteristics. Because of the small size of weapon objects and the distance from the CCTV, the weapons were partially or fully concealed in many frames. In similar research of occluded object detection, Wang et al. [57] created a dataset with nine occlusion levels over two dimensions. Researchers distinguished three degrees of object occlusion: FG-L1 (20–40%), FG-L2 (40–60%), and FG-L3 (60–80%). Furthermore, they established three degrees of context occlusion surrounding the object: BG-L1: 0–20%, BG-L2: 20–40%, and BG-L3: 40–60%. The authors manually segregated the occlusion images into two groups: mild occlusion (two sub-levels) and strong occlusion (three sub-levels), with increasing occlusion levels. Their study investigated the challenge of recognizing partly occluded items under occlusion and discovered that traditional deep learning algorithms that mix proposal networks with classification networks can not recognize partially occluded objects robustly. The experimental results show that this issue can be resolved.

As far as small objects detection is concerned, the image quality and image size should be the focus but also needs to keep all information of CCTV visions. Seo et al. [58] used the traditional object recognition network in the proposed method. The extremely low resolution recognition problem deals with images with a resolution lower than 16 × 16. The authors created a high resolution (HR) image set utilizing the original images from the CIFAR datasets, which have a resolution of 32 × 32 pixels. The original CIFAR datasets were downsampled to 8 × 8 pixels to create the low resolution (LR) image set and then the bilinear approach to upsample the HR pictures to 32 × 32 pixels. The authors utilized the ImageNet 32 × 32 pixels subset available in downsampled ImageNet to employ the same parameters as those used for the CIFAR datasets instead of the original ImageNet images with resolutions higher than 224 × 224 pixels. In final conclusions, authors compared the accuracy of object recognition on the various LR images. The results indicated a drastic decrease in recognition accuracy for the LR input images, which implies the difficulty of the given task. Furthermore, Courtrai et al. [59] used the quality of the super-resolved image to improve small image detection, due to the increasing size and quantity of Enhanced Deep Residual Super-resolution’s (EDSR) residual blocks and the addition of an auxiliary network to aid in better localizing items of interest during the training phase. Consequently, compared to the original EDSR framework, detection performance had been markedly improved. In conclusion, the modified number of residual blocks of 32 × 32 and 96 × 96 enables detection correctness even on images with extremely low spatial resolution, provided that a high quality image of the same objects was available. From these proposed approaches, the higher quality images fed to networks enable higher correctness to the detection model.

When comparing the ratio of labels per image and the average sizes of the labels from each dataset, in the Weapon Detection Dataset [43], the number of labels was slightly higher than the images in the dataset, when looking at the ratio of labels per image of 1.15, which means a single image had at least one annotation with an average label size of 261 × 202 pixels, indicating medium size object labels. For the Weapon Detection System [44], the number of labels was two time higher than the images in the dataset, when looking at the ratio of labels per image of 1.82, a single image had at least one annotation and was more likely to have more than one annotation each, with an average label size of 150 × 116 pixels indicating medium- to small-sized object labels. Even though the label sizes of the Weapon Detection Dataset [43] and Weapon Detection System [44] contain higher resolution of their object images, the field we were interested in was covering CCTV visioning, where the Mock Attack Dataset [55] provided an answer to our objectives. However, looking over the Mock Attack Dataset [55], the number of labels was two times lower than the images in the dataset with a huge number of object imbalances. When looking at the ratio of labels per image of each class, they were all lower than one, which means a single image might have no annotation at all, but an average label size of 56 × 99 to 40 × 50 pixels indicated small-sized object labels that we were interested in for this research. In our newly collected quality Armed CCTV Footage (ACF) dataset, the number of labels resembled an image in the dataset, when looking at the ratio of labels per image of each class, they were slightly higher than one, which means a single image contained at least one annotation, but with an average label size of 49 × 62 to 43 × 66 pixels indicating small-sized object labels.

## 3. ACF Dataset

To increase the efficiency of the criminal intent detection system, an artificial intelligence platform was used to detect crime intent. CCTV footage was recorded with a video recorder, and each frame of the footage was utilized to teach the deep learning model to detect the probability of criminal actions. The system detects weapons in the image using weapon detection and then evaluates the results of each procedure and determines that the individual in the event has a high potential of committing a crime.

### 3.1. CCTV Cameras Setting

In this experiment, data for crime intention were collected using images of criminals armed with pistols and knives. The authors collected the data by using Hikvision CCTV cameras that captured full high-definition (HD) video, which were 1920 × 1080 pixels at 30 frames per second (fps). The two cameras were placed at opposite corners of the staging area and were facing each other. The cameras were installed on 2.8 m tall stands with the camera angle facing down approximately 30–35 degrees to the ground, as shown in Figure 3.

### 3.2. Data Collection

The datasets were taken in controlled scenarios of the following crime-intended event. In Figure 4, the armed event contains four scenarios in which in the first scenario (**a**) the offender runs straight from left to right. In the second scenario, (**b**) the offender runs straight from top to bottom. For the third scenario, (**c**) the offender runs diagonally from bottom left to top right, and finally, (**d**) the offender walks in a spiral pattern from outside toward the center. The following scenarios were created to control the visibility of the weapon object in the image. The different directions of movement presented different weapon angles to the cameras. For creating the weapon detection system for CCTV surveillance, the dataset for creating the detection model needed to be informative.

The weapons used in the footage were two combat knives and three different types of BB guns. The combat knives were primarily dissimilar by color, black and white. The BB guns used different models, which were BB gun airsoft Galaxy G.26 (replica of Sig 226), BB gun airsoft Galaxy G.052 (replica of M92), BB gun airsoft Galaxy G29 (replica of Makarov), as shown in Figure 5.

The data collection took place in the Faculty of Engineering, Mahidol University, and the regulations were followed, which required the authors to hold faculty permits that allowed them to take armed weapon scenarios footage. Upon the permission and security personnel approved, the activity received a valid duration for data collection. The cameras were setup as mentioned in Section 3.1. The data collection consisted of two CCTV cameras located at different locations, outdoor and indoor, in the same area, covering three different areas from the parking lot and one corridor, and different scenarios enacted. The offenders were role played by four male participants, aged 24–36 years old, who alternately carried a weapon in each round under morning- and afternoon-simulated lighting environments. Each round of data collection contained up to two offenders running in the direction following each scenario to increase the number of weapons visible in frame, as illustrated in Figure 6.

The location of data collection for the camera setup was composed of four different areas around the Faculty of Engineering to create different backgrounds, different lighting between indoors and outdoors, and lighting at different times of the day, thus creating a variety of CCTV image characteristics that were suitable for weapon detection, as shown in Figure 5. The CCTV cameras were in standby mode to record the overall mocking scenarios of carrying a pistol or knife during the permitted duration. The overall duration for usable data collection took 2 h 18 min and 2 s using the two cameras, and each location had to be set up to collect both pistol and knife armed scenarios, as shown in Table 2.

Parking lot 1: Parking lot 1 was located at the corner of the Faculty of Engineering parking lot, where the area had no shadows of surrounding objects, such as trees or buildings, casting on the footage area. The area’s light environment was totally dependent on the weather, which was sunny. The data collection at parking lot 1 took 38 min 44 s for both the pistol and knife data collection. The data were collected during the morning and afternoon of the same day. Three participants were alternately armed with a weapon following data collection scenarios, and each scenario may have had up to two weapon-carrying participants.Parking lot 2: Parking lot 2 was located at the edge of the Faculty of Engineering parking lot where the area had shadows from buildings and trees casting on the footage area. The data collection at parking lot 2 took 28 min 8 s for both the pistol and knife data collection, where the dataset was collected during the morning and afternoon of the same day. Three participants were alternately armed with a weapon following the data collection scenarios. Each scenario may have had up to two participants carrying weapons.Parking lot 3: Parking lot 3 was located at the edge of the Faculty of Engineering parking lot connected to a grass field where the area had the shadow of nearby trees casting on the footage area. The data collection at parking lot 3 took 45 min 35 s of both the pistol and knife data collection, and the data were collected during the afternoon. Four participants were alternately armed with weapons following the data collection scenarios. Each scenario may have had up to two participants carrying weapons.Corridor: The corridor was located at the corridor connected to the classrooms on the second floor of the Faculty of Engineering where the area had the facade’s shadow casting on the footage area. The data collection at the building corridor took 25 min 35 s for both the pistol and knife data collection, where the data were collected in the afternoon. Two participants were alternately armed with weapons following the data collection’s scenarios. Each scenario may have had up to two participants carrying weapons.

### 3.3. Data Acquisition

The CCTV footage was collected from the controlled scenarios mentioned in Section 3.2. The authors collected the data by using Hikvision CCTV cameras that captured full HD videos, in MP4 format, and the JPEG images were then extracted from the video footage frames. The total duration of data collection footage from the two CCTV cameras took 2 h 18 min and 2 s. The images were selected from the footage of 1 fps, achieving a total of 8319 images. Then, the dataset was partitioned into Dataset 1: ACF_Pistol Dataset, Dataset 2: ACF_Knife Dataset, and Dataset 3: ACF Dataset, to provide resources for object detection of different classes and to visualize how the data of each class impacted each method used in our research. The datasets were allocated to the following dataset:Dataset 1: ACF_Pistol Dataset

The ACF_Pistol Dataset contained 4426 CCTV images of pedestrians armed with a pistol. The images in this dataset depicted a person holding a pistol, which were full HD images with a resolution of 1920 × 1080 pixels and various scene compositions.

2.Dataset 2: ACF_Knife Dataset

The ACF_Knife Dataset contained 3559 CCTV images of pedestrians armed with a knife. The images included were a knife held by a person, which were full HD images with a resolution of 1920 × 1080 pixels and various scene compositions.

3.Dataset 3: ACF Dataset

The Pistol and Knife Datasets were the combined dataset of Dataset 1 and Dataset 2, comprising 8319 CCTV images of pedestrians armed with a pistol and knife.

The partition datasets were divided depending on their class to determine how well each class, both single class (pistol or knife) or multiple class (pistol and knife), can perform in selected object detection and preprocessing methods. Furthermore, the partition datasets could be used for other related research.

## 4. Methodology

For the overall method, the CCTV footage was extracted into full HD Armed CCTV images as mentioned in Section 3.3. The datasets were preprocessed using tiling methods, then tiled images were fed into the weapon detection deep learning model, and the results were evaluated, as shown in Figure 7.

### 4.1. Preprocessing

As the images were fed into the deep learning network, the input size of the model was of concern. The higher dimension of the input size to be compatible with the input image size required higher computation resources and time, while using a smaller input size, the input image was compressed and lost its object characteristic. The image tiling method was used to divide the image into smaller images along with the label file to keep computational costs low while maintaining performance comparable to CNNs that scale and analyze a single image [49,50,51], where use of image tiling approach is mentioned. Unel et al. [53] can also greatly improve performance using the tiling approach, increasing the mAP from 11% to 36%, while the precision is increased by 2.5× and 5×, respectively. The tiling method was more effective on small items while also improving the detection of medium-sized items. The number of bounding box labels increased compared to the default dataset due to the overlap part of the bounding box where the cut seam is divided, as shown in Figure 8.

Therefore, the tile number must be of interest. Unel et al.’s [53] experiment on tiling images in different sizes feed to the network, and the authors suggest avoiding using a number of tiles higher than 3 × 2 because the inference time can increase significantly while the accuracy remains unchanged. The 3 × 2 tiling image has a better outcome within the available processing limits, while the 5 × 3 tiling image reports slightly higher accuracy with a significant computation overhead, compared to our method where the tiling image was in 2 × 2 tiles, as the image size should not be smaller than 512 × 512 pixels, the input size of the network, and it took up acceptable computation time compared to the higher split tiles.

The tiling images reduced the image size, while it increased the ratio of weapon annotation to the image size, as shown in Figure 9. The label size to image size ratio of the raw image was 0.0014, presenting the weapons as small objects, while the tiling increased the ratio to 0.0056, which was four times the raw image annotations.

The label files were labeled by LabelImg [60] created by Tzutalin. LabelImg is an open-source program under MIT license that allows a developer to draw bounding boxes on images to indicate objects of interest. In this research, the images were divided into 2 × 2 tiles. The original size image was then divided into four images of 960 × 540 pixels. After preprocessing with the image tiling method, the data contained in each dataset were composed of: Tiling Dataset 1: A total of 17,704 tiling images of Dataset 1, which contained images of a person holding a pistol in different scenarios.Tiling Dataset 2: A total of 14,236 tiling images of Dataset 2, which contained images of a person armed with a knife.Tiling Dataset 3: The combined dataset of Tiling Dataset 1 and Tiling Dataset 2, which comprised a total of 33,276 tiling images of Dataset 3, which contained images of a person armed with a pistol or knife.

### 4.2. Weapon Detection Model Training

In our research, the weapon detection model training was trialed as two methods. The first method was meant to directly feed an image to the object detection network for teaching the model to recognize the interested weapon. In the second method, images from the ACF Dataset underwent a preprocessing method called images tiling, and then, the tiled images were fed into the network, as shown in Figure 7. Both methods used the object detection model in TensorFlow Model Zoo [61].

The CCTV footage was extracted into images to create the image dataset for object detection training. The image dataset was then preprocessed using the image tiling method. The datasets were then fed to the object detection training model for the detection of weapons in the image. The field of interesting in the detected results contained a pistol and knife.

### 4.3. Evaluation Metrics

Annotated datasets were available for training and evaluating models in popular competitions such as Common Objects in Context (COCO) [62], PASCAL visual object classes (VOC) [63], and Open Images Dataset [64]. The object detection method’s assessment metrics included a baseline performance analysis to compare the model’s performance. The assessment was based on the TensorFlow Object Detection API’s standard loss function for prediction class classification and bounding box localization [65]. Padilla et al. [66] gives an overview of the object detection competition assessment techniques and looks at how different implementations might impact the evaluation outcomes.

Intersection over Union (IoU)

IoU is a bounding box measure of the overlap between the ground truth and the detection result:(1)$$\mathrm{IoU}=\frac{Area\;of\;Overlap}{Area\;of\;Union}=\frac{area\left(B_{p}\;\cap \;B_{g}\right)}{area\left(B_{p}\;\cup \;B_{g}\right)},$$
where $B_{p}$ is the predicted bounding box, and $B_{g}$ is the ground truth bounding box. 

An IoU threshold is usually expressed as a percentage, and commonly used thresholds are 50 and 75 percent. According to the metrics used for object detection evaluation, true positives (TP) are cases in which the confidence value is greater than a certain threshold of the matched class result, and the bounding box of the prediction has an IoU value greater than the threshold. When multiple predictions replicate the same ground truth, only the prediction with the highest confidence score is referred to as a true positive, while the others are referred to as false positives (FP). When the discovered ground-confidence truth’s score is less than the threshold, the detection is termed a false negative (FN). Conversely, a true negative (TN) is defined when the confidence value of a non-interesting detection is lower than the threshold. Many researchers [67,68,69,70] utilized IoU losses to evaluate the degree of accuracy of network prediction and balanced L1 losses to increase the localization accuracy even further. Rezatofighi et al. [68] addressed the drawbacks of IoUs by proposing a generalized loss and then implemented this generalized IoU (GIoU) metric as a loss in existing object identification systems.

2.Precision

When the value of false positives is high, precision is concerned. Many test results would be false positives if the model had poor precision. Precision is used to calculate the number of positive class outcomes.

3.Recall

The recall is helpful when the cost of false negatives is high. Recall indicates the false negatives against true positive ones. False negatives are particularly relevant for the prevention of disease detection and other safety-related predictions. It estimates the positive class predictions from all positive instances from the dataset.
(2)$$\mathrm{Recall}=\frac{\mathrm{TP}}{\mathrm{TP}+\mathrm{FN}\;}$$

4.Average Precision (AP)

The metric for evaluating object recognition was created as part of the PASCAL VOC [38] competition, whose authors chose to use the AP metric for the competition. The average precision is a single data point metric that encapsulates 0 to 1 of a precision and recall value. It was specified by the author as the region under the curve obtained by precision and recall sampling at eleven discrete recall values [0.0:1.1:0.1].

In 2012, the VOC competition iteration [71] made a minor adjustment to the AP calculation by sampling precision and recall at all unique recall intervals; instead of the 11 predefined recall intervals, all points are referred to as average precision. This approach effectively benchmarks the algorithm for object detection against low scores for AP. The region under the precision–recall curve can be determined as AP since all recall points are now included.

5.Average Recall (AR)

AR is a method to assess object detector confidence for a given class. AR uses the recall values that acquired thresholds in the interval [0.5, 1] to showcase proposal performance across IoU thresholds. An IoU of 0.5 reflects an object’s approximate localization and is the least acceptable IoU by most standards, whereas an IoU of 1 represents the identified object’s actual position. According to Hosang et al. [72], enhancing proposal localization accuracy is as crucial as improving recall for object detection. Average recall benefits both high sensitivity and specificity and decent localization and corresponds remarkably well with detection performance.

6.Mean Average Precision (mAP)

Regardless of the interpolation method, average precision is determined for each class individually. In large datasets with specific categories, having a distinctive measure that can indicate the accuracy of detections in all classes is beneficial. Within those cases, the mAP, which is essentially the average *AP* throughout all categories, is measured.
(3)$$\mathrm{mAP}=\frac{1}{C\;}\overset{C}{\underset{i=0}{\sum }}AP_{i}$$
where $AP_{i}$ is the AP value for the *i*th class and *C* is the total number of classes.

Without changing the network architectures, Chen et al. [73] studied performance gains in cutting-edge one-stage detectors based on AP loss over various types of classification losses on various benchmarks.

## 5. Experimental Results

### 5.1. Experimental Setup

The object detection model training required information or image patterns. The dataset needed to be properly prepared and contained both the image files and the label files, indicating the object of weapon in the images. The labeled datasets were then split into two sets: training and testing. The partitioned dataset was required to be converted into a TFRecord file. This was used for training the object detection model in the TensorFlow Model Zoo [61].

This research utilized a TensorFlow model from the Model Zoo, which included:(1)SSD MobileNet V2;(2)EfficientDet D0;(3)Faster R-CNN Inception Resnet V2.

During the model training procedure, the TFRecord files were utilized to pass image information across the network. The configuration pipeline file was used to tune the model by changing the variables in each iteration to avoid overtraining the object detection model. The configuration pipeline setup for object detection in each architecture is shown in Table 3. The labelmap file necessitated to additionally define the type of objects to be used for the model learning. When testing the weapon detection model, the value of the results provided the position of the detected weapon on the images as well as the classified object type. The pre-trained model configuration pipeline mostly used the default value provided by the TensorFlow Model Zoo examples where we adjusted the training steps and batch size to fit the data in our dataset. While the activation function and optimizer selection were from the default value. The input size of each model was chosen to match its model’s architecture.

The authors divided the dataset into a training set and a test set in an 80:20 ratio, with the 80 percent sectioned into a training set and the remaining 20 percent used as a validation set. For testing the model, the test dataset contained 660 images of the Dataset 1 and Dataset 2, while Dataset 3 contained the combined Dataset 1 and 2 test datasets. The test dataset was used to evaluate the model for practical use as well as to evaluate model inferences. The tiling test dataset was preprocessed from the previous test dataset, which now contained the 2640 images of Tiling Dataset 1 and Tiling Dataset 2, while Tiling Dataset 3 contained the combined Tiling Dataset 1 test dataset and Tiling Dataset 2 test dataset, a total of 5280 images, as shown in Table 4.

### 5.2. Weapon Detection on ACF Dataset 

For the weapon detection experiment, the self-collected datasets were used, composed of Dataset 1, Dataset 2, and Dataset 3. The pre-trained model from the TensorFlow framework was used for the weapon detection training:SSD MobileNet V2;EfficientDet D0;Faster R-CNN Inception Resnet V2.

From Table 5, Dataset 1, which contained 4426 images, was used to train the weapon detection models, and the 885 images were utilized for the evaluation. SSD MobileNet V2 models had the highest mAP value of 0.427, mAP at the 0.5 threshold of IoU (0.5 IoU) equal to 0.777, and mAP at the 0.75 threshold of IoU (0.75 IoU) equal to 0.426. While on Dataset 2 comprising 3559 images, 711 images were used for evaluation. The highest mAP value was equal to 0.544, with mAP at 0.5 IoU equal to 0.907, and mAP value at 0.75 IoU equal to 0.585 from SSD MobileNet V2. Lastly, Dataset 3 comprising 8319 images, of which 1664 images were used for evaluation, achieved the highest mAP value at 0.495, with mAP at 0.5 IoU equal to 0.861, and mAP value at 0.75 IoU equal to 0.517 from SSD MobileNet V2. For the overall experiment, the SSD MobileNet V2 yielded the best evaluation result in every dataset.

In Figure 10, the receiver operating characteristic curve (ROC curve) of (**a**) Dataset 1, (**b**) Dataset 2, and (**c**) Dataset 3 are illustrated. The ROC curve was plotted from the true positive rate to the false-positive rate, which represented the classification performance of the model. The area under curve (AUC) was calculated from the ROC curve to evaluate the performance, an AUC value nearest to 1 indicated better classification performance. From Table 6, the AUC value of the SSD MobileNet V2 performed less well in the classification task, in contrast to its significant localization, while EfficientDet D0 and Faster R-CNN Inception Resnet V2 classified effectively.

Figure 11 and Figure 12 show the results of object detection using Dataset 1, Dataset 2, and Dataset 3, respectively. The example images in Figure 8 and Figure 9 are the results from the SSD MobileNet V2, which achieved the highest mAP value in both 0.5 and 0.75 IoU thresholds. The orange arrows in the images indicate the weapon objects in the image, with the green rectangle indicating the predicted bounding box of the weapon class for single-class weapon detection and the green and blue rectangles indicating the predicted results of the object detection model (pistol and knife) for multi-class weapon detection. The probability of detecting a weapon was higher if it is close to the camera.

From Table 7, Tiling Dataset 1, containing 14,164 images, was used to train the weapon detection models, with 3540 images being used for evaluation. The models achieved the highest mAP value of 0.559, the mAP value at 0.5 IoU was equal to 0.900, and the mAP value at 0.75 IoU was equal to 0.636 from SSD MobileNet V2. For Tiling Dataset 2, comprising 11,392 images, 2844 images were used for evaluation. The highest mAP value was equal to 0.630, with mAP at 0.5 IoU equal to 0.938, and mAP at 0.75 IoU equal to 0.747 from Faster R-CNN Inception Resnet V2. Lastly, Tiling Dataset 3, comprising 26,620 images, 6656 images were used for evaluation and to achieve the highest mAP value at 0.544, with mAP at 0.5 IoU equal to 0.891, and mAP at 0.75 IoU equal to 0.616 from SSD MobileNet V2. For the overall experiment, the SSD MobileNet V2 yielded the best evaluation result in every dataset.

For the overall results using the Tiling ACF Dataset, the proposed object detection models achieved the best experimental results compared with the experimental results from the image tiling. The results clearly confirm that we created a capable object detection model using the image tiling method for weapon detection from CCTV footage.

From Figure 13, the ROC curve of (**a**) Tiling Dataset 1, (**b**) Tiling Dataset 2, and (**c**) Tiling Dataset 3 on the image tiling method are illustrated. The ROC curve was plotted from the true positive rate to the false-positive rate, which represents the classification performance of the model. The AUC value was calculated from the ROC curve to evaluate the performance. The AUC value closest to 1 indicated better classification performance. From Table 8, the AUC values of SSD MobileNet V2 of the overall experiment performed less well in the classification task, in contrast to its effective localization, while EfficientDet D0 and Faster R-CNN Inception Resnet V2 can classify effectively. Compared to Table 8, the AUC values of the tiling approach are slightly lower, indicating that the image tiling method produces a higher false-positive rate than the usual approach.

Figure 14 and Figure 15 show results of object detection with the image tiling method using Tiling Dataset 1, Tiling Dataset 2 and, and Tiling Dataset 3, respectively. The example images in Figure 11 are the results from the SSD MobileNet V2 that achieved the highest mAP in both 0.5 and 0.75 IoU thresholds. The orange arrows in the images indicate the ground truth of the objects of interest, while the green rectangles indicate the pistols predicted by the object detection model. Compared to Figure 12, the small pistol is more likely to be detected when using the tiling method. The comparison of the evaluation of the test dataset is discussed in the next section. Figure 15 shows that the SSD MobileNet V2 can accurately detect and predict a pistol and a knife.

From Table 9, the weapon detection models were tested with the ACF test dataset, which was not included in the model training comprising 660 images of Dataset 1 and Dataset 2 and 1320 images of Dataset 3. For Dataset 1, the highest mAP value at 0.5 threshold IoU achieved 0.667. In Dataset 2, the SSD MobileNet V2 achieved the highest mAP value at 0.5 threshold IoU of 0.789. Lastly, Dataset 3 gave the highest mAP value at 0.5 threshold IoU of 0.717 by SSD MobileNet V2. For the overall dataset, the SSD MobileNet V2 yielded the best result.

The weapon detection models were tested with the Tiling ACF test dataset comprising 2640 images of Tiling Dataset 1 and Tiling Dataset 2, and 5280 images of Tiling Dataset 3. For Tiling Dataset 1, the highest mAP value at 0.5 threshold IoU achieved 0.777. In Tiling Dataset 2, the SSD MobileNet V2 also achieved the highest mAP value at 0.5 threshold IoU of 0.779. Lastly, Tiling Dataset 3 gave the highest mAP value at 0.5 threshold IoU of 0.758 by SSD MobileNet V2. For the overall dataset, SSD MobileNet V2 yielded the best result.

Based on the evaluation of the weapon detection model, the MobileNet V2 gave higher detection precision than EfficientDet D0 and Faster R-CNN Inception Resnet V2 overall, while in Dataset 2, the EfficientDet D0 used tiling enhance performance of weapon detection, but the SSD MobileNet V2 realized a higher mAP than Tiling EfficientDet D0. As mentioned in the previous section, the example images in Figure 11 and Figure 12 show a non-detected knife, whereas Dataset 2 itself can lead to more false-positive prediction because the model shape and color can easily merge with the background. Tiling Dataset 1 and Tiling Dataset 2 achieved the highest mAP of 0.777 and 0.779 on SSD MobileNet V2, respectively, while Tiling Dataset 3 achieved slightly lower mAP of 0.758 on the same architecture. However, with two classes detection, Tiling Dataset 3 was more promising to further integrate into the system.

As for Table 10, the weapon detection models were tested with the same default test dataset to evaluate the inference of the model, comprising 660 images of Dataset 1 and Dataset 2 and 1320 images of Dataset 3. For the lightweight model, the SSD MobileNet V2 on Dataset 1 and Dataset 2, the inference time taken was 43 to 44 milliseconds, and the throughput was 16 images per second, while in Dataset 3, the inference time was decreased to 41 milliseconds and achieved a throughput at nine images per second. For the EfficientDet D0 for all datasets, the inference time taken was 46 to 47 milliseconds and the throughput was 15 images per second for Dataset 1 and Dataset 2, while the throughput of 8 images per second was achieved for Dataset 3.

While a massive size model such as Faster R-CNN Inception Resnet V2 achieved an inference time at 274–294 milliseconds and achieved a throughput of Dataset 1 at 3 images per second, there was a throughput of Dataset 2 at 2 images per second and a throughput of Dataset 3 at 1 image per second.

Compared with the tiling method dataset, SSD MobileNet V2, the inference time was increased at two times more than the default ACF Dataset, and the throughput decreased at 7 images per second. The inference time of EfficientDet D0 was increased to 110 milliseconds, and the throughput decreased to 6 images per second on Tiling Dataset 1 and Tiling Dataset 2. While Tiling Dataset 3 performed a throughput of 3 images per second. Lastly, the inference time of Faster R-CNN Inception Resnet V2 was increased to 1060 milliseconds, and the throughput decreased to 1 image per second on Tiling Dataset 1 and Tiling Dataset 2, while Tiling Dataset 3 performed a throughput of 0.3 images per second.

From all the experiments, the use of the tiling to process can significantly enhance detection accuracy while preserving performance comparable to CNNs that scale and analyze a single image. Daye [67] stated that rather than compressing an image to network size, image tiles are given to the network so that the image retains its original quality. As a result, high-ratio scaling has no effect on the image, and small objects keep their original dimensions. Image tilling, however, can lengthen inference time and add extra post-processing overheads for merging and refining predictions. The image tiles achieved from the image tiling method were concerned with the input size of the image to feed into the network. The smaller input size of our selected model was 512 × 512, and the images should be larger than the input size of the network. When feeding images into the network, the deeper the layer, the less shiftable its feature maps [74,75,76,77]. When using a smaller size of tile images than the input size, the image is compressed and loses more object characteristics; therefore, the tile number must be included. Unel et al. [53] experimented on tiling images in different sizes feeding to 304 × 304 pixels of input size to the network. The authors avoided using a number of tiles higher than 3 × 2 because the inference time increases significantly while the accuracy remains unchanged. The 5 × 3 tiling image has the better outcome within the available processing limits, while the 3 × 2 tiling image has the highest accuracy with a significant computation overhead. This was compared to our method that used the tiling image of 2 × 2 tiles as the image size, was not smaller than 512 × 512 pixels for the input size of the network, and took up acceptable computation times compared to higher split tiles.

The architecture that can effectively perform the overall task was MobileNet V2. From the results of the previous section, the tiling approach of MobileNet V2 improved the localization task in all experiments. Furthermore, in the classification task, the overall results show that the tiling method decreases the correctness of the classification weapon. However, as each architecture has lower classification performance, EfficientDet D0 shows a slight decrease in classification results when compared to Table 9 and Table 11. Finally, the processing time is acceptable when compared to the lightest architecture such as SSD MobileNet V2, which requires the fastest inference time, as shown in Table 10.

### 5.3. Approach Comparison with Related Research

The Mock Attack Dataset from González et al. [45] is a Full HD image with the dimension of 1920 × 1080 pixels and consists of 5149 images, but only the weapon labels consist of 2722 labels of three weapon types: handgun, short rifle, and knife. The dataset is divided into a training set and a test set of ratios of 80:20, which consist of 2378 and 544 labels, respectively. For the image tiling method, each image and bounding boxes are decomposed into 2 × 2 tiled images. The dataset now contains a total of 3106 labels, divided into 2485 labels of the training set and 621 labels of the test set.

According to González et al. [45], the Mock Attack Dataset was trained using Faster RCNN Resnet 50 from the TensorFlow Model ZOO object detection API with 40,000 steps and an initial learning rate of 0.002. The random horizontal flip augmented method was used during the training. Table 11 shows that the model achieved a mAP of 0.009, a mAP of 0.5 IoU threshold of 0.034, and a mAP of 0.75 IoU threshold of 0.003. While using our method with the image tiling method on the same detector, Faster R-CNN, the object detection models are trained on Faster R-CNN Inception Resnet V2 and evaluated to return a bounding box indicating the location of the object in the image and classifying the weapon type. The model was able to achieve a mAP of 0.077, a mAP of 0.5 IoU threshold of 0.192, and a mAP of 0.75 IoU threshold of 0.035. The enhancement of our methods was compared to González et al. [45], who achieved mAP 8.56 times better, the mAP of IoU threshold at 0.5 achieved 5.65 times better, and the mAP of IoU threshold at 0.75 achieved 11.67 times better.

After acquiring results of the Tiling Mock Attack Dataset from González et al. [45], the author tested the methods with different models, including EfficientDet D0 and SSD MobileNet V2, following our experiments. Table 12 shows the concluded results of each model. The overall result of the tiling methods shows the improvement of the model evaluation compared to the non-tiling method, where the EfficientDet D0 model performs better on the overall image with a mAP of 0.092 and a 0.5 IoU threshold of mAP of 0.254. The enhancement of our methods compared to González et al. [45] has achieved mAP 10.22 times better, the mAP of IoU threshold at 0.5 achieved 7.47 times better, and the mAP of IoU threshold at 0.75 achieved 19.33 times better.

## 6. Discussion

The experiments evaluated the result of the object detection model on pistols and knives using a self-collected dataset of CCTV footage demonstrating that the ACF Dataset can perform better on SSD MobileNet V2. Tiling Dataset 1 achieved the highest mAP of 0.777, while the Tiling Dataset 3 on the same architecture achieved a lower mAP of 0.758. However, since two classes were detected, Tiling Dataset 3 appears to be more suitable for further integration into the CCTV system.

### 6.1. Importance of ACF Creation

As the different environments might affect the model’s performance [56], the detector was biased toward variables such as poor contrast, headlights, and specular reflection conditions. Because of the small size of the weapon objects and the distance from the CCTV, the weapons were partially or fully concealed in many frames. However, for small weapon detection, the image quality was low, the performance of the detector should be focused but also needs to keep all CCTV vision information [58,59], and the higher quality images feeding to networks enable higher correctness of the detection model.

Quality datasets are very important for research on weapon detection from CCTV footage. For the overall results using the ACF Dataset, the proposed object detection models achieved the best experimental results compared with the experimental results from González et al. [45]. Due to the small number of images and inadequate class of public datasets for our interested domain in weapon detection, we chose to create an ACF Dataset for our proposed method. The ACF Dataset has a total of 8319 CCTV images of pedestrians armed with pistols and knives.

### 6.2. Importance of Tiling Approach

Furthermore, the image tiling method was applied to the mock attack dataset [45]. As mentioned in Section 5.3, the method can enhance the efficiency of detecting weapons on images. When training a deep learning object detection, several hundred images of different types were needed to boost the efficiency of the detection process. Due to the following public dataset containing only 2722 annotations with three weapon categories, this is a small amount. Even though the evaluation value is not high, the mock attack dataset itself has many empty annotation images and imbalances in annotation between the three weapon types.

In similar research of occluded object detection [57], the challenge of recognizing items under occlusion and discovering that traditional deep learning algorithms that mix proposal networks with classification networks cannot recognize partially occluded objects robustly, but the experimental results showed that occluded weapon detection can be resolved. While our weapon labels also included some partial occluded weapon objects, the detector was able to recognize more occluded weapons in the Tiling ACF Dataset, where the tiling methods helped in increasing partial weapon object recognition.

Therefore, the image tilling method can lengthen the inference time and adds extra processing overhead for merging and refining the predictions. The Tiling Dataset 3 using SSD MobileNet V2 achieved the highest mAP of 0.758, which performed better than Dataset 3 using SSD MobileNet V2, achieving 0.717.

## 7. Conclusions and Future Works

### 7.1. Conclusions

This study aimed to analyze the ACF Dataset for utilizing the appropriate training data for weapon detection to enhance the small weapon detection efficiency on CCTV footage, and to develop a weapon detection model using image tilling-based deep learning. Object detection models based on the Model Zoo Object Detection API were used to identify weapons in an image using SSD MobileNet V2, Faster R-CNN Inception Resnet V2, and EfficientDet D0. The self-collected ACF Dataset was collected with CCTV cameras in different scenarios and locations where weapon detection can be significantly enhanced. Furthermore, the results clearly confirm that the image tiling method helps enhance the efficiency of detecting small weapons on CCTV footage images. On SSD MobileNet V2, Tiling Dataset 3 achieved the highest mAP of 0.758 compared to the raw CCTV image evaluation method, achieving 0.717. This image tiling method can be used in various applications for a small region of an image object.

When using the proposed tiling approach on a public dataset as mentioned in Section 5.3, the mock attack dataset [45] was trained and provided better evaluation results than the raw mock attack images detection training, which achieved the highest mAP of 0.092 on the EfficientDet D0 architecture. The tiling approach achieved a significantly better mAP of 10.22 times, the mAP of IoU threshold at 0.5 achieved 7.47 times better, and the mAP of IoU threshold at 0.75 achieved 19.33 times better. The result is quite low compared to our training with the Tiling ACF Dataset. The ACF Dataset is informative enough for enhancing the efficiency of weapon detection on real CCTV footage, and the proposed tiling method on the ACF Dataset can additionally enhance the performance of weapon detection, while using tiling approaches on tiling test datasets can significantly improve the false-negative rate, as illustrated in Figure 14 and Figure 15.

### 7.2. Limitations and Future Work

Our quality ACF Dataset can enhance weapon detection on CCTV footage, but as for our limitations, the dataset was collected on knife and pistol weapon images during the daytime where the weather was sunny and cloudy. Detection might produce wrong detection results when there is inference in different environments, night light images, other types of weapons or objects that neglect our weapon collections. Although our dataset enhances the performance of weapon detection, the errors of detection such as false positives still occurred, and the examples of error detection are further demonstrated in Appendix A. From our evaluation experiment in Table 10, the image tiling method can increase the inference time up to two times compared to the conventional image detection using SSD MobileNet V2. While the throughput is decreased, it is capable of images inference at 7 images per second. Real-time weapon detection might not be available using this proposed method.

For further development of our work, we will emphasize the generalization of our approach by incorporating additional data augmentation techniques to address the constraints. To reduce false negatives to the system, occluded weapons should be included. Video detection should be used to indicate weapon visibility in certain event durations, adding variation of a test dataset that includes non-weapon objects such as a wallet, phone, watch, bottle, book, and glass can visualize prediction errors caused by the inability to distinguish classes of similar objects. In addition, we will emphasize the weapon detection technique, YOLO architecture, to increase the model accuracy and to increase the model inference time speed for the ability to infer real-time detection either on video detection or CCTV Streaming Real Time Streaming Protocol (RTSP) IP render streaming.

## Figures and Tables

**Figure 1 sensors-22-07158-f001:**
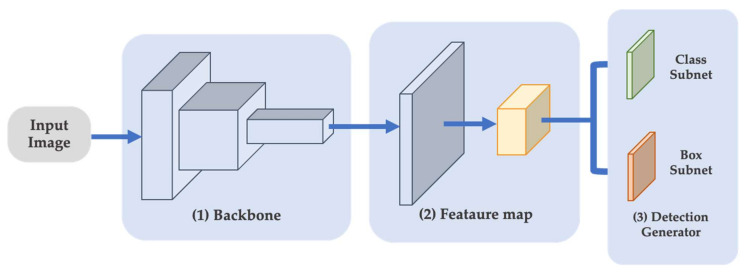
One-stage object detection architecture.

**Figure 2 sensors-22-07158-f002:**
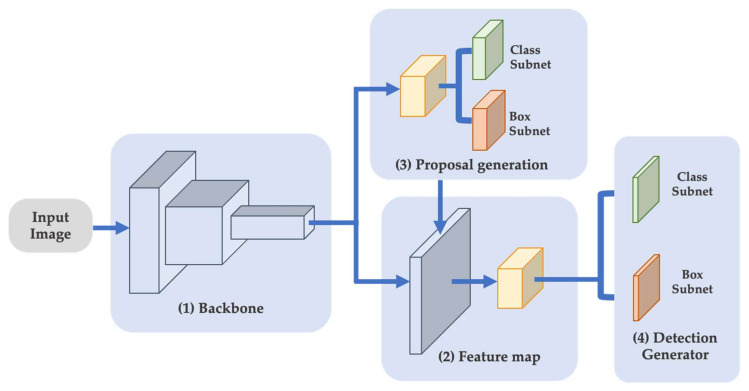
Two-stage object detection architecture.

**Figure 3 sensors-22-07158-f003:**
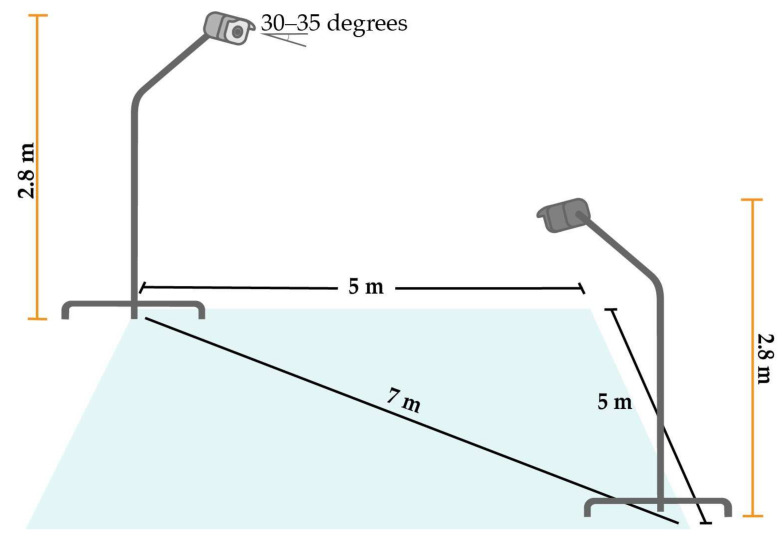
CCTV camera setup.

**Figure 4 sensors-22-07158-f004:**
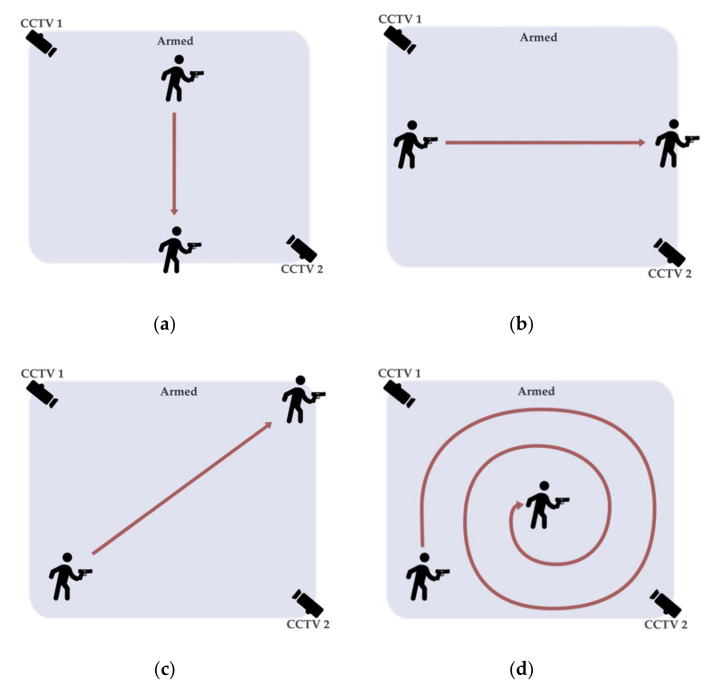
Scenarios where the offender runs out of the frame in three directions: (**a**) left to right, (**b**) top to bottom, (**c**) bottom left to top right, and remains in frame in (**d**) the spiral pattern.

**Figure 5 sensors-22-07158-f005:**
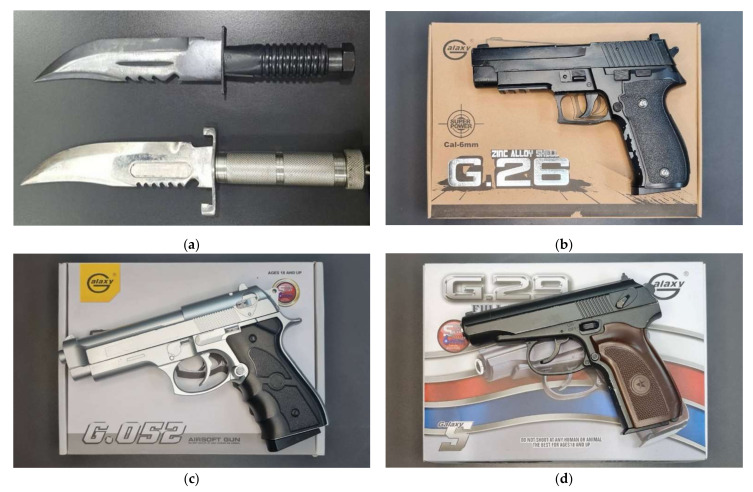
Example of weapons used in the data collection process: (**a**) combat knives, (**b**) BB gun airsoft Galaxy G.26 (Sig 226), (**c**) BB gun airsoft Galaxy G.052 (M92), (**d**) BB gun airsoft Galaxy G29 (Makarov).

**Figure 6 sensors-22-07158-f006:**
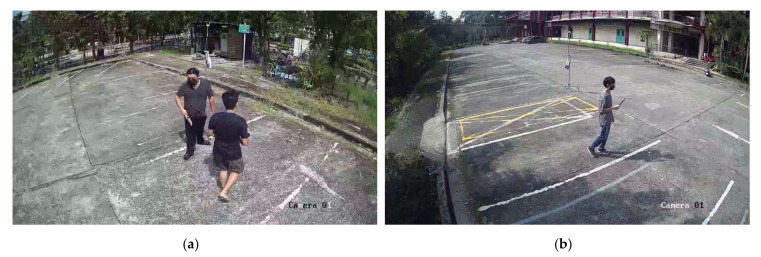
Example of location in the data collection process: (**a**) Parking lot 1, (**b**) Parking lot 2, (**c**) Parking lot 3, (**d**) Corridor.

**Figure 7 sensors-22-07158-f007:**
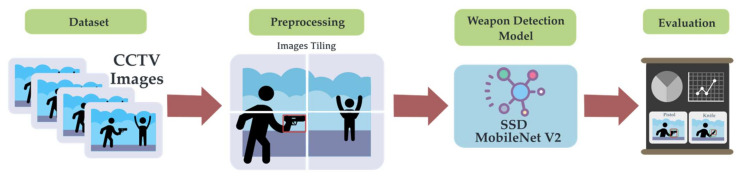
Methodology for weapon detection.

**Figure 8 sensors-22-07158-f008:**
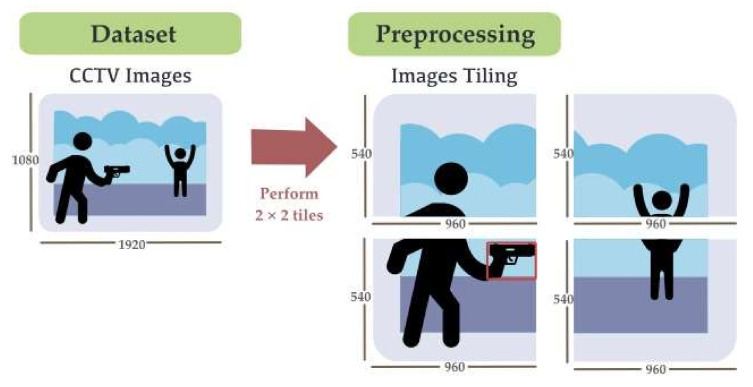
Image tiling method.

**Figure 9 sensors-22-07158-f009:**
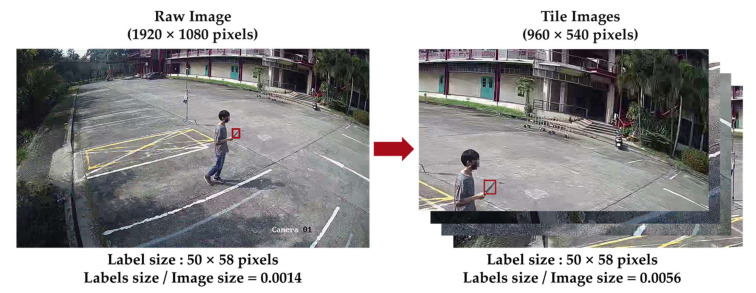
Example of raw image implementation on tiling methods.

**Figure 10 sensors-22-07158-f010:**
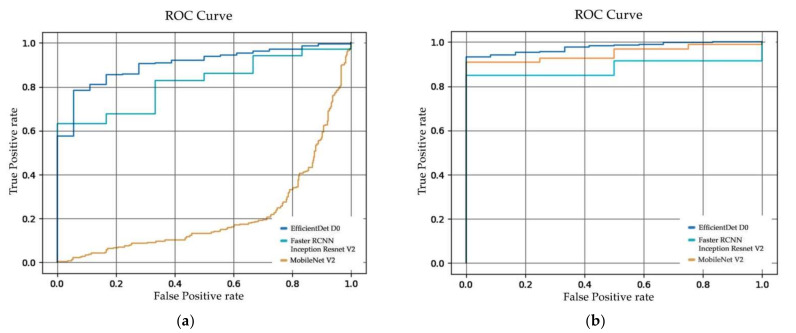
ROC curves of SSD MobileNet V2, EfficientDet D0, and Faster R-CNN Inception Resnet V2 from (**a**) Dataset 1, (**b**) Dataset 2 and (**c**) Dataset 3.

**Figure 11 sensors-22-07158-f011:**
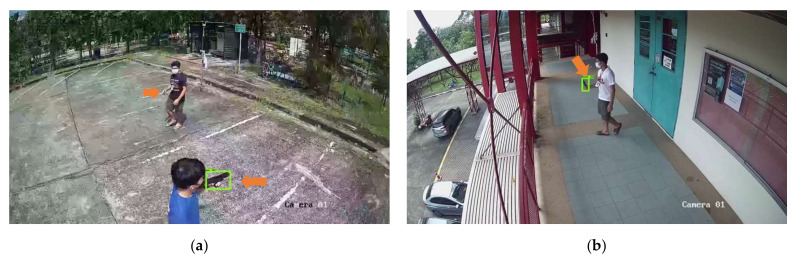
SSD MobileNet V2 model prediction on example images using (**a**) Test Dataset 1 and (**b**) Test Dataset 2.

**Figure 12 sensors-22-07158-f012:**
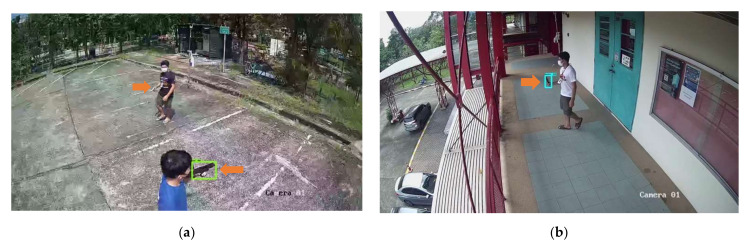
SSD MobileNet V2 model prediction on example images using Test Dataset 3 of (**a**) the same image of Test Dataset 1 and (**b**) the same image of Test Dataset 2.

**Figure 13 sensors-22-07158-f013:**
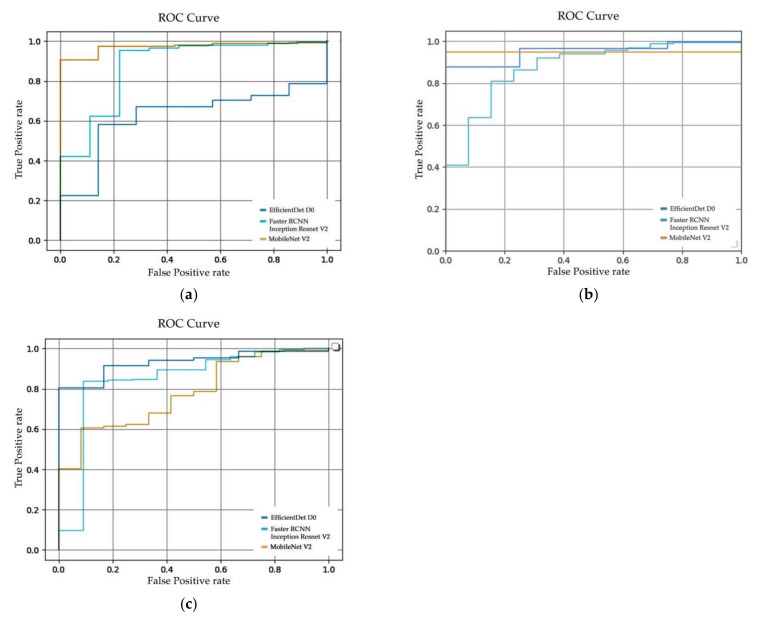
ROC curves of SSD MobileNet V2, EfficientDet D0, and Faster R-CNN Inception Resnet V2 from (**a**) Tiling Dataset 1, (**b**) Tiling Dataset 2 and (**c**) Tiling Dataset 3.

**Figure 14 sensors-22-07158-f014:**
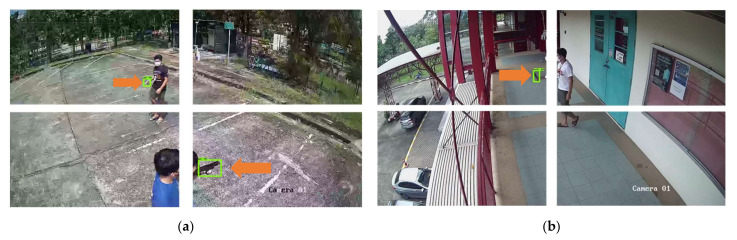
SSD MobileNet V2 model prediction on example images using (**a**) Tiling Test Dataset 1 and (**b**) Tiling Test Dataset 2.

**Figure 15 sensors-22-07158-f015:**
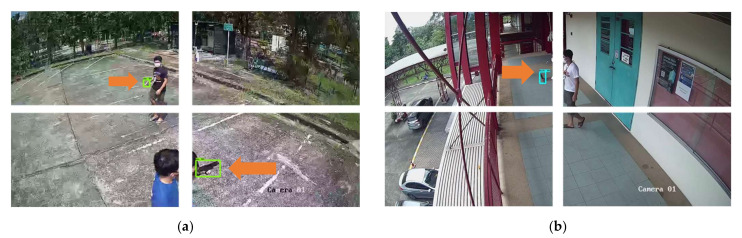
SSD MobileNet V2 model prediction on example images using Tiling Test Dataset 3 of (**a**) the same image of Tiling Test Dataset 1 and (**b**) the same image of Tiling Test Dataset 2.

**Table 1 sensors-22-07158-t001:** Public dataset information.

Dataset	Image Size	Class	Number of Images	Labels	Ratio (Labels/Image)	Avg. Pixels
Weapon detection dataset [43]	240 × 145 pixels–4272 × 2848 pixels	Gun	3000	3464	1.15:1	(261 × 202)
Weapon detection system [44]	99 × 93 pixels–6016 × 4016 pixels	Gun	4940	9202	1.82:1	(150 × 116)
Mock Attack Dataset [55]	1920 × 1080 pixels	Handgun	5149	1714	0.32:1	(40 × 50)
		Short rifle		797	0.15:1	(56 × 99)
		Knife		210	0.04:1	(40 × 52)
ACF Dataset (Ours)	1920 × 1080 pixels	Pistol	8319	4961	1.12:1	(49 × 62)
		Knife		3618	1.02:1	(43 × 66)

**Table 2 sensors-22-07158-t002:** Data collection information.

Location	Video Duration	Weapon	Time ^1^	Participant ^2^
Parking lot 1	38 min 44 s	Pistol Knife	Morning Afternoon	3
Parking lot 2	28 min 8 s	Pistol Knife	Morning Afternoon	3
Parking lot 3	45 min 35 s	Pistol Knife	Afternoon	4
Corridor	25 min 35 s	Pistol Knife	Afternoon	2

^1^ Time indicates daytime of data collection where Morning was 5:00–11:59 a.m. and Afternoon was 12:00 a.m.–17:59 p.m. ^2^ Number of participants shown in each location footage.

**Table 3 sensors-22-07158-t003:** Model configuration information.

Architecture	SSD MobileNet V2	EfficientDet D0	Faster R-CNN Inception Resnet V2
Input size	640 × 640	512 × 512	640 × 640
Activation function	RELU_6	Swish	SOFTMAX
Steps	30,000	30,000	30,000
Batch size	20	12	2
Evaluation metrics	COCO	COCO	COCO
Optimizer	Momentum optimizer	Momentum optimizer	Momentum optimizer
Initial Learning rate	0.04	0.04	0.04
Training time	6 h 21 min 28 s	13 h 8 min 13 s	3 h 49 min 0 s

**Table 4 sensors-22-07158-t004:** Dataset information.

Dataset	Training Set	Validation Set	Total	Test Dataset ^1^
Dataset 1: ACF_Pistol Dataset	3541	885	4426	660
Dataset 2: ACF_Knife Dataset	2848	711	3559	660
Dataset 3: ACF Dataset	6655	1664	8319	1320
Tiling Dataset 1: Tiling ACF_Pistol Dataset	14,164	3540	17,704	2640
Tiling Dataset 2: Tiling ACF_Knife Dataset	11,392	2844	14,236	2640
Tiling Dataset 3: Tiling ACF Dataset	26,620	6656	33,276	5280

^1^ Test dataset containing unseen CCTV footage for unseen image evaluation.

**Table 5 sensors-22-07158-t005:** The object detection evaluation on the ACF Dataset.

Type	Architecture	mAP	0.5 IoU	0.75 IoU
Dataset 1	SSD MobileNet V2	0.427	0.777	0.426
	EfficientDet D0	0.296	0.763	0.154
	Faster R-CNN Inception Resnet V2	0.279	0.686	0.145
Dataset 2	SSD MobileNet V2	0.544	0.907	0.585
	EfficientDet D0	0.347	0.804	0.223
	Faster R-CNN Inception Resnet V2	0.350	0.734	0.261
Dataset 3	SSD MobileNet V2	0.495	0.861	0.517
	EfficientDet D0	0.242	0.661	0.111
	Faster R-CNN Inception Resnet V2	0.376	0.804	0.275

**Table 6 sensors-22-07158-t006:** AUC of ROC curve.

Dataset	Architecture	AUC
Dataset 1	SSD MobileNet V2	0.215
	EfficientDet D0	0.903
	Faster R-CNN Inception Resnet V2	0.819
Dataset 2	SSD MobileNet V2	0.949
	EfficientDet D0	0.977
	Faster R-CNN Inception Resnet V2	0.883
Dataset 3	SSD MobileNet V2	0.832
	EfficientDet D0	0.913
	Faster R-CNN Inception Resnet V2	0.884

**Table 7 sensors-22-07158-t007:** The object detection evaluation on the Tiling ACF Dataset.

Dataset	Architecture	mAP	0.5 IoU	0.75 IoU
Tiling Dataset 1	SSD MobileNet V2	0.559	0.900	0.636
	EfficientDet D0	0.303	0.726	0.202
	Faster R-CNN Inception Resnet V2	0.445	0.870	0.391
Tiling Dataset 2	SSD MobileNet V2	0.256	0.689	0.113
	EfficientDet D0	0.488	0.855	0.526
	Faster R-CNN Inception Resnet V2	0.630	0.938	0.747
Tiling Dataset 3	SSD MobileNet V2	0.544	0.891	0.616
	EfficientDet D0	0.419	0.819	0.386
	Faster R-CNN Inception Resnet V2	0.343	0.751	0.247

**Table 8 sensors-22-07158-t008:** AUC of ROC curve on image tiling method.

Dataset	Architecture	AUC
Tiling Dataset 1	SSD MobileNet V2	0.972
	EfficientDet D0	0.623
	Faster R-CNN Inception Resnet V2	0.876
Tiling Dataset 2	SSD MobileNet V2	0.953
	EfficientDet D0	0.952
	Faster R-CNN Inception Resnet V2	0.886
Tiling Dataset 3	SSD MobileNet V2	0.779
	EfficientDet D0	0.931
	Faster R-CNN Inception Resnet V2	0.845

**Table 9 sensors-22-07158-t009:** Evaluated model on the test dataset.

Dataset	Architecture	0.5 IoU	
		Raw Image	Tile Image
Tiling Dataset 1	SSD MobileNet V2	0.667	0.777
	EfficientDet D0	0.547	0.507
	Faster R-CNN Inception Resnet V2	0.534	0.673
Tiling Dataset 2	SSD MobileNet V2	0.789	0.779
	EfficientDet D0	0.704	0.719
	Faster R-CNN Inception Resnet V2	0.534	0.654
Tiling Dataset 3	SSD MobileNet V2	0.717	0.758
	EfficientDet D0	0.545	0.632
	Faster R-CNN Inception Resnet V2	0.620	0.620

**Table 10 sensors-22-07158-t010:** Evaluated model inferences on the test dataset.

Dataset	Architecture	Raw Image		Tile Image	
		Inference	Throughput	Inference	Throughput
Dataset 1	SSD MobileNet V2	44.3 ms	16 images/s	98.7 ms	7 images/s
	EfficientDet D0	46.4 ms	15 images/s	112.8 ms	6 images/s
	Faster R-CNN Inception Resnet V2	277.1 ms	3 images/s	1061.3 ms	1 images/s
Dataset 2	SSD MobileNet V2	43.2 ms	16 images/s	96.5 ms	7 images/s
	EfficientDet D0	47.9 ms	15 images/s	110.6 ms	6 images/s
	Faster R-CNN Inception Resnet V2	293.7 ms	2 images/s	1064.3 ms	1 images/s
Dataset 3	SSD MobileNet V2	41.2 ms	9 images/s	94.6 ms	4 images/s
	EfficientDet D0	46.7 ms	8 images/s	112 ms	3 images/s
	Faster R-CNN Inception Resnet V2	274.1 ms	1 images/s	1066.5 ms	0.3 images/s

**Table 11 sensors-22-07158-t011:** The object detection evaluation on the mock attack dataset.

Method	mAP	0.5 IoU	0.75 IoU
Mock Attack Dataset (González et al. [45])	0.009	0.034	0.003
Tiling Mock Attack Dataset (Ours)	0.077	0.192	0.035

**Table 12 sensors-22-07158-t012:** The object detection evaluation on the Tiling Mock Attack Dataset.

Architecture	mAP	0.5 IoU	0.75 IoU
EfficientDet D0	0.092	0.254	0.028
Faster R-CNN Inception Resnet V2	0.077	0.192	0.035
SSD MobileNet V2	0.084	0.200	0.051

## Data Availability

The Armed CCTV Footage dataset and source code are available at https://github.com/iCUBE-Laboratory/The-Armed-CCTV-Footage (accessed on 31 August 2022).

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
