# Peer review of "ACF: An Armed CCTV Footage Dataset for Enhancing Weapon Detection"

_sensors, 2022, doi:10.3390/s22197158_

Round 1
Reviewer 1 Report
The authors designed a CCTV-based dataset to evaluate weapon detection algorithms in this manuscript. The dataset contains 7981 images, including knife and pistol. They have captured their images using two Hikvision CCTV cameras with 1920 x 1080 resolution. They have divided their dataset into training and validation sets to evaluate deep learning-based weapon detection algorithms by maintaining an 80:20 ratio. In the current manuscript version, authors have claimed that the SSD MobileNet V2 model outperformed to detect weapons in the proposed dataset with mAP of 0.777.
Here are some suggestions for improvements:
1. While automated annotation tools are available in the literature, it is unclear why authors manually annotated their ACF dataset.
2. It is unclear which compression technique was used to save the captured images.
3. The dataset contains only 7981 images which seems to be a small dataset to feed a neural network. It would be best if the author created a video dataset for weapon detection.
4. A very limited number of weapons were considered in this dataset, including pistols and knives. There should be more variations of weapons introduced by the author.
5. For the above reasons, I believe the proposed manuscript is not ready for publication, especially in a journal.
Author Response
Dear reviewers,
We would like to thank all of you for your constructive comments in this round of review. Your comments provided valuable insights to refine the content and analysis.
We have carefully proofread our manuscript and made the corrections in the sincere hope that it meets your requirements.
Please find enclosed feedback on the changes we made to the manuscript according to your feedback.

Reviewer 2 Report
Dear Authors,
Thank you for your work on image analysis.
The paper seems a divulgation one but faces a few technical questions.
The critical part is section 4, which I suggest you rewrite entirely, putting in evidence the decision you take and why you take that specific decision.
Unlike the rest, I can read this section with difficulty as it seems to me that it is not well organized.
1) You face the problem by using two different kinds of preprocessing. Why?
2) During the second type of preprocessing, you split the image into 4 areas. What happens if the weapon is divided among the splitter images?
3) You use more models provided by TensorFlow Model ZOO, but it is not clear how you combine these algorithms with the preprocessing you propose.
4) The conclusion seems disconnected from the rest of the paper. Please rewrite this section putting in evidence the material you have previously described.
This is only a suggestion, you can freely adopt another strategy. Your paper contains a lot of information, perhaps you could divide it into the shortest paragraphs having each one a subtitle or a sub-subtitle that you can refer to in the following parts of the paper.
Sorry if I write you again, rewrite this paragraph. It is not clear.
Regards
Author Response
Dear reviewers,
We would like to thank all of you for your constructive comments in this round of review. Your comments provided valuable insights to refine the content and analysis.
Please find enclosed feedback on the changes we made to the manuscript according to your feedback.

Reviewer 3 Report
-
Summary:
Due to the small weapon objects in the footage, the authors collected the self-collected mockup CCTV footage of armed pedestrians composed of pistol and knife armed in different scenarios. Different deep models for object detection have been validated on the newly collected dataset.
-
Strengths:
i) the authors collected a new dataset for weapon detection.
ii) Different deep models have been tested for pistol and knife detection on the newly collected dataset.
-
Weaknesses:
i) My biggest concern comes from the fact that there are verbose descriptions of the common sense (such as the various metrics, activation function), while lack of information about the details of the newly collected data.
ii) The authors mentioned several challenges for detecting weapons from CCTV footage, such as low-resolution image, small in size, occlusion and different scenarios, but it is unclear how the newly collected data can help alleviate those challenges.
-
Rating and Justification:
Based on the two main concerns as above, I made my final recommendation.
-
Additional comments:
i) the literature is way too verbose. The authors are suggested to cite and discuss the related works only, while removing the introduction of some common knowledge, such as the sliding window, activation functions, evaluation metrics (shorten, and move to experimental section), etc.
ii) Except the fact that the size of the newly collected dataset is several hundreds larger than the previous datasets, what makes this dataset unique? How can it help alleviate the challenges for weapon detections in CCTV footage?
iii) Please add more details about the data collection, such as how many participants, data collection period (does it contain both daytime and nighttime CCTV footage?)Are participants carrying other objects? How is the data from two CCTVs used?
iv) The conclusions and future works are unclear, the authors are suggested to revise the content.
v) What is the computational cost for the proposed method? Can the model run in real-time?
Author Response

(The authors gave the same response as above.)

Round 2
Reviewer 2 Report
Dear Authors,
Thank you for your work. Please share your dataset as you wrote and share your source code. After sharing, I will accept the paper.
Regards.
Author Response
Dear reviewer,
We would like to thank you for your constructive comments in this round of review. Your comments provided valuable insights to refine the content and analysis. Please find enclosed feedback on the changes we made to the manuscript according to your feedback.
Reply Review# (Reviewer 2)
|
Technical Comments |
|
|
01 |
Thank you for your work. Please share your dataset as you wrote and share your source code. After sharing, I will accept the paper. |
|
Answer |
We would like to thank you again for your constructive comments. We have rewritten Section 2 (Line: 951-952). The Armed CCTV Footage dataset and source code are available at https://github.com/iCUBE-Laboratory/The-Armed-CCTV-Footage. |
Reviewer 3 Report
Thank the authors, most of my concerns have been resolved.
The paper is re-organized. More related works have been cited and discussed, more details are added about the dataset collection as well.
But, there are still some comments for the revised manuscript:
-
As commented before, it is unclear how this newly collected dataset can help alleviate the challenges for weapon detections in CCTV footage, such as low-resolution image, small in size, occlusion and different scenarios, etc. Please clearly justify this new dataset.
-
It may be better to detect weapons from short video than from static images. For example, a weapon may be occluded by body parts in one frame, but not in the next few frames.
-
More diverse objects, including both weapons and other non-weapons, are expected to be used during data collection. For example, will the system recognize a non-weapon object as a weapon, and then send a false-positive alert, and in a real world setting, this can be a big issue if the system frequently recognizes non-weapon objects as a weapon.
Author Response
Dear reviewer,
We would like to thank you for your constructive comments in this round of review. Your comments provided valuable insights to refine the content and analysis. Please find enclosed feedback on the changes we made to the manuscript according to your feedback.
Reply Review# (Reviewer 3)
|
Technical Comments |
|
|
01 |
As commented before, it is unclear how this newly collected dataset can help alleviate the challenges for weapon detections in CCTV footage, such as low-resolution image, small in size, occlusion and different scenarios, etc. Please clearly justify this new dataset. |
|
Answer |
Thank you for the suggestion. We have rewritten Section 2.2 (Line: 272-282, 293-325, 334-337), Section 6.1 (Line: 851-858), and Section 6.2 (Line: 876-886) to provide more explanation.
|
|
02 |
It may be better to detect weapons from short video than from static images. For example, a weapon may be occluded by body parts in one frame, but not in the next few frames. |
|
Answer |
As the suggestion, we have added the weapon detection results on sequence images using ACF dataset in Appendix A (Line: 958-970). The model was inference on short video sequence images.
|
|
03 |
More diverse objects, including both weapons and other non-weapons, are expected to be used during data collection. For example, will the system recognize a non-weapon object as a weapon, and then send a false-positive alert, and in a real world setting, this can be a big issue if the system frequently recognizes non-weapon objects as a weapon. |
|
Answer |
As the suggestion, we have added the limitations and future works in Section 7.2 as shown in Line 914-935.
|